# The association between arthritis and cognitive function impairment in the older adults: Based on the NHANES 2011–2014

Taihong Lv[1☯], Hanming Yu[2☯], Zishuo Ji[3], Li Ma[1]*

**1** Department of General Practice, Beijing TianTan Hospital, Capital Medical University, Beijing, China, **2** Department of Pulmonary and Critical Care Medicine, The Fourth Affiliated Hospital of China Medical University, Shenyang, China, **3** Department of Neurology, Beijing TianTan Hospital, Capital Medical University, Beijing, China

☯ These authors contributed equally to this work.
* mali_ttyy@126.com

## Abstract

### Objective

Arthritis has been postulated as a prevalent potential risk factor for the emergence of dementia and cognitive impairment. This conjecture prompted an examination of the correlation between arthritis and cognitive impairment using the National Health and Nutrition Examination Survey (NHANES) repository. The analysis was meticulously adjusted for potential confounders such as age and assorted systemic comorbidities, to ensure robustness in the results obtained.

### Methods

Among 2,398 adults aged 60 years and above, logistic regression and cubic spline models were employed to elucidate the relationship between arthritis and cognitive performance. This was assessed utilizing tests such as Immediate Recall test (IRT), Delayed Recall test (DRT), Animal Fluency Test (AFT), and Digit Symbol Substitution Test (DSST).

### Results

In our investigation, a total of 19931 individuals were analyzed, among which 2,398 patients (12.03%) were identified with arthritis. Subjects with arthritis inflammation had lower DSST and AFT scores compared to the healthy group, indicating cognitive decline. After adjusting for all covariates, arthritis was significantly associated with higher DSST and AFT scores by logistic regression modeling (OR: 0.796, 95% CI: 0.649–0.975; OR: 0.769, 95% CI: 0.611–0.968).

### Conclusion

Our analysis underscores the potential linkage between arthritis prevalence and cognitive impairment within a nationally representative of US older adults.

**Data Availability Statement:** The dataset central to this study was obtained from the National Health and Nutrition Examination Survey (NHANES), which is accessible to the public via the official

NHANES website: https://www.cdc.gov/nchs/nhanes/index.htm.

**Funding:** The author(s) received no specific funding for this work.

**Competing interests:** The authors have declared that no competing interests exist.

**Abbreviations:** AFT, Animal Fluency Test; BMI, Body mass index; CERAD, Consortium to Establish a Registry for Alzheimer's Disease; CI, Confidence interval; DRT, Delayed Recall test; DSST, Digit Symbol Substitution Test; IRT, Immediate Recall test; MCI, Mild cognitive impairment; NHANES, National Health and Nutrition Examination Survey; OR, Odds Ratio; RA, Rheumatoid arthritis.

# 1 Introduction

In the United States, a study indicated that nearly one in three individuals aged 65 years and above were affected by dementia or mild cognitive impairment (MCI) [1]. The progressive decline in cognitive capabilities results in impairment of intellect, behavior, and function, significantly disrupting daily activities. Koller and Bynum report an adjusted national dementia prevalence of 8.24%, with state-specific rates ranging between 5.96% and 9.55% [2]. Presently, dementia ranks as one of the leading causes of mortality worldwide, and its associated death toll is projected to rise in conjunction with increases in population size and aging demographics [3,4]. It is estimated that the global incidence of dementia will surge from 57.4 million cases (95% uncertainty interval: 50.4–65.1) in 2019, to 152.8 million cases (uncertainty interval: 130.8–175.9) by the year 2050 [5].

Arthritis represents a diverse array of joint-related pathologies marked by inflammation, commonly associated with pain and the potential for progressive joint degeneration [6]. Rheumatoid arthritis (RA) and osteoarthritis stand as the primary categories of arthritis [7]. From 1990 to 2017, there has been an observed increase of 7.4% in the global age-standardized prevalence, and an 8.2% rise in the incidence of rheumatoid arthritis, resulting in 246.6 and 14.9 cases per 100,000 individuals, respectively [8]. Rheumatoid arthritis extends its impact beyond joint deterioration—it is implicated in conditions such as vasculitis, malignancies, pulmonary and cardiovascular diseases, as well as mental health disorders, often leading to enduring harm [9]. The prevalence of extra-articular manifestations of RA varies from 17.8% to 40.9%, with potential involvement of multiple organ systems, inclusive of neurological and psychiatric domains [10,11]. Moreover, individuals diagnosed with rheumatoid arthritis exhibit an elevated risk of developing dementia when compared to the wider population [12], a finding that is consistent across multiple studies [13–15]. Osteoarthritis (OA) is a multifaceted and progressive chronic condition, presently afflicting an estimated 250 million individuals globally [16]. There is a recognized correlation between OA and an elevated risk of dementia and cognitive impairment (CIM), suggesting that efficacious interventions targeting OA could potentially diminish the emergence of new dementia or CIM cases [17]. Moreover, research indicates a strong association between the psychological symptoms and cognitive patterns with OA-induced pain [18,19].

In summation, the mechanisms that underpin the relationship between arthritis and cognitive deterioration remain predominantly enigmatic. Hypothesized contributory factors include chronic inflammation [20,21], alterations within the immune system [22], and incessant pain and discomfort [23,24]. Future investigative efforts necessitate the rigorous scientific development and pragmatic application of mechanistic research, as the current evidence base is inadequate to substantiate any single explanatory mechanism conclusively.

Presently, there is a discernible dearth of research employing the NHANES database to examine the nexus between arthritis and cognitive impairment. Addressing this void, our current cross-sectional investigation leveraged NHANES data spanning 2011 to 2014 to probe into the potential link between arthritis prevalence and cognitive impairment incidence among the older adults in the United States. We posited that arthritis-afflicted individuals would demonstrate an increased susceptibility to cognitive impairment.

# 2 Materials and methods

## 2.1. Ethics statement

In accordance with the protocol sanctioned by the National Center for Health Statistics Research Ethics Review Board, all participants submitted written informed consent prior to

their inclusion in the NHANES. The dataset utilized in this research is de-identified and publicly available, thereby negating the requirement for additional ethical approval from the Swedish Ethical Review Authority. This approach is consistent with established national guidelines governing research conduct.

## 2.2. Study population

Fig 1 illustrates the methodological framework, participant selection, and exclusion criteria employed within this investigation. The study harnessed two widely available datasets from the NHANES for the 2011–2014 survey cycles. Selection of participants for each cycle adhered to a structured criterion: 1) exclusion of individuals younger than 20 years of age; 2) removal of participants without cognitive function tests; 3) discussion of subjects without an arthritis diagnosis; 4) elimination of entries deficient in data across a breadth of covariates, encompassing age, gender, ethnicity, educational level, marital status, income, tobacco usage, alcohol consumption, physical activity, sleep disorders, hypertension, and diabetes. Collectively, across the two cycles, the study incorporated a composite cohort of 2398 participants for analysis.

## 2.3. Cognitive impairment

The cohort was comprised solely of individuals aged 60 and above. Cognitive function was characterized across four distinct constructs. The Consortium to Establish a Registry for Alzheimer's Disease (CERAD) Word Learning Test gauged the capacity for new verbal information acquisition, encapsulating both Immediate Recall test (IRT) and Delayed Recall test (DRT). To assess executive function, the Animal Fluency Test (AFT) measured categorical verbal fluency. Meanwhile, the Digit Symbol Substitution Test (DSST) evaluated attributes such as processing velocity, consistent attentiveness, and working memory capacity. An aggregate cognitive impairment index was formulated by summing the metrics across these dimensions, where the lowest quartile received a score of 1 while remaining quartiles were allocated a score of 0 [25]. Consequently, escalating scores were indicative of intensified cognitive deterioration.

## 2.4. Covariates

Building upon the findings from extant scholarly work, a comprehensive array of potential covariates linked to arthritis has been methodically documented within the academic discourse [26,27]. To ascertain the intervening effects among these covariates, our analysis was underpinned by the methodological construct proposed by Baron and Kenny [24]. We scrutinized an array of covariates encapsulating sociodemographic factors, behavior patterns, and health-related attributes.

From a sociodemographic stance, the investigations included age categories (60–69, 70–80), gender (female, male), ethnic composition (non-Hispanic white, Mexican American, non-Hispanic black, other/multiracial), marital status (married, widowed, divorced, separated, never married, cohabiting), educational level (below high school, high school graduate and above), and financial standing gauged by the Poverty Income Ratio (PIR)—defining poverty threshold as a PIR below 1.

Behavioral attributes were quantified based on smoking status (individuals with a history of 100+ cigarettes were characterized as smokers; responses recorded as "yes" or "no"), alcohol consumption ("no" or "yes"), and physical activity level ("inactive" or "active").

Health-related vectors encompassed body mass index (BMI under 24 signifying underweight/normal; BMI of 24 or above denoting overweight/obesity), sleep disorders (none or confirmed), incidence of hypertension (none or confirmed), and diabetes mellitus status

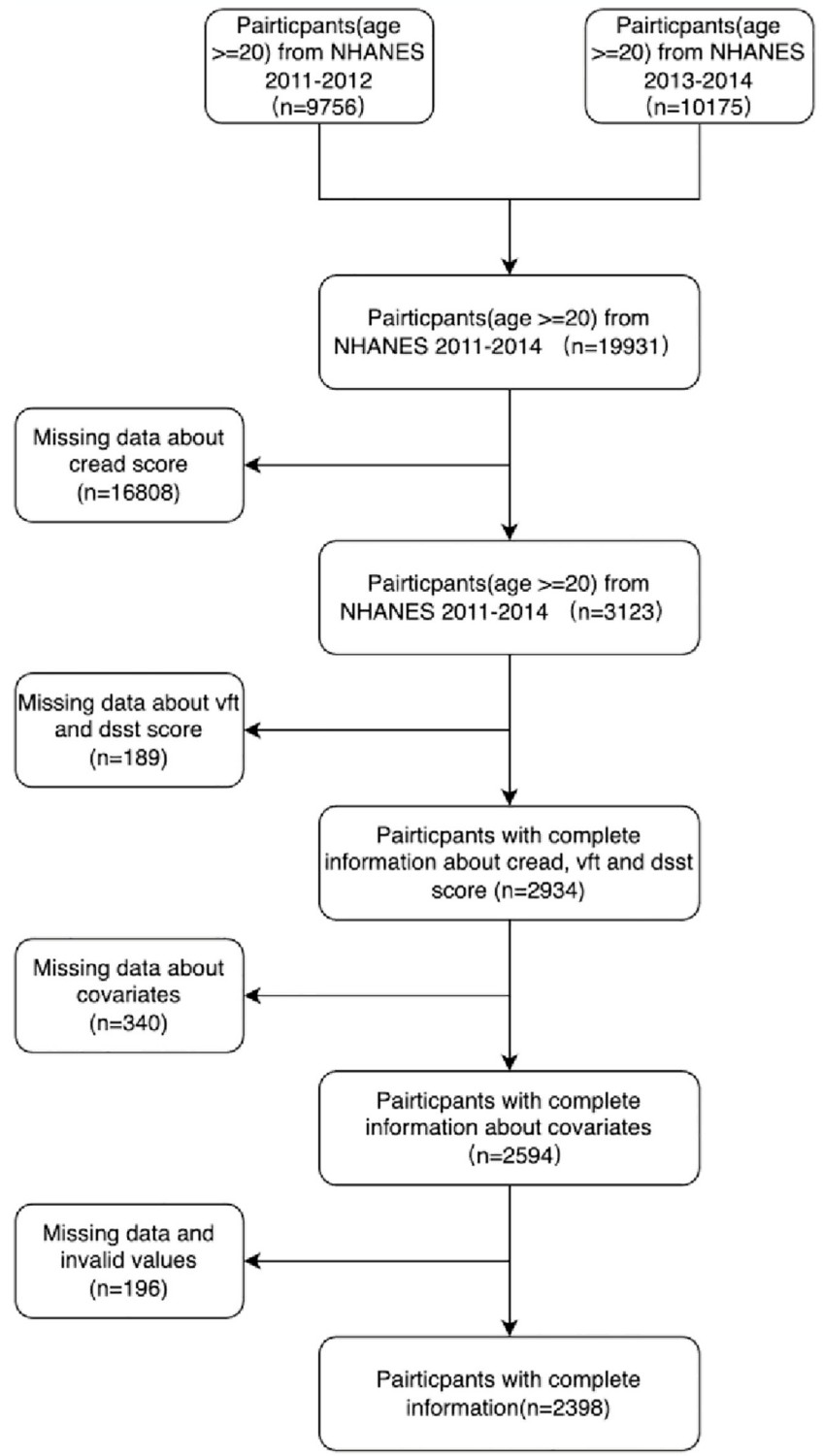

**Fig 1. Flowchart of the population included in our final analysis.**

(none or confirmed). Each element was meticulously examined for its potential mediating role in the intricate interplay between arthritis and cognitive impairment.

## 2.5. Statistical analysis

In this study, a cross-sectional methodology was employed to analyze data from adult participants in the National Health and Nutrition Examination Survey (NHANES) covering the period from 2011 to 2014. The aim was to investigate the association between arthritis and cognitive decline in the older population. The dataset was meticulously stratified to include a wide range of demographic variables such as age, gender, and ethnicity, ensuring that the analysis accounted for the diversity within the population and helped in identifying specific subgroups that might be more affected by arthritis-related cognitive decline. Linear regression models were utilized to examine the relationship between arthritis and cognitive impairment, allowing for the assessment of how cognitive scores (dependent variable) varied with the presence of arthritis (independent variable), while controlling for other covariates. Model 1 included basic demographic adjustments, Model 2 incorporated additional health-related variables, and Model 3 included all potential confounders such as socioeconomic status and lifestyle factors. Following the initial linear regression analysis, multivariate logistic regression was conducted to further explore the relationship between arthritis and cognitive impairment, providing a more comprehensive understanding by considering the impact of multiple factors simultaneously. The dependent variable was the presence or absence of cognitive impairment, and the independent variables included arthritis status along with other covariates like age, gender, ethnicity, education level, and comorbid conditions. Odds Ratios (OR) and 95% Confidence Intervals (CI) were calculated to determine the strength and significance of the associations. The statistical significance of the results was determined using p-values and confidence intervals, with associations having p-values less than 0.05 considered statistically significant, indicating a less than 5% probability that the observed associations were due to chance.

## 3 Results

Table 1 delineates the baseline demographics of the study participants over successive survey iterations. The incidence of arthritis in the older adults, aged over 60 years, was documented at 49.3%. Notably, there was a discernible increment from 47.2% during the 2011–2012 interval to 51.2% in the 2013–2014 timeframe; nevertheless, this rise did not attain statistical significance (p = 0.054). Furthermore, the variations in the prevalence of cognitive impairment, as quantified by AFT and DSST cognitive scores, failed to manifest statistical significance (p = 0.209, p = 0.391, respectively). Conversely, the prevalence discrepancies in cognitive impairment, as determined by CERAD assessment, were found to be statistically significant (p < 0.001).

Table 2 exhibits the attributes of the cohort with and without cognitive deficits as measured by the three distinct scales. According to the univariate analysis, factors such as smoking status, BMI, and sleep disturbances bore no correlation with the incidence of cognitive impairment. In contrast, advanced age (70–80 years), female gender, racial background, income level, alcohol intake, educational attainment, marital status, physical activity frequency, hypertension, and diabetes mellitus emerged as potential determinants of cognitive impairment among American adults.

Table 3 delineates the outcomes of a linear regression analysis assessing the impact of arthritis on cognitive decline, utilizing three distinct evaluation methods. Post extensive adjustment for a breadth of sociodemographic, behavioral, and health-related covariates. For

Table 1. Characteristics of participants across NHANES 2011–2014cycles.

| Variables | Classification | N | Year | | Total | p |
|---|---|---|---|---|---|---|
| | | | 2011–2012 | 2013–2014 | | |
| arthritis | no | n | 590 | 625 | 1215 | 0.054 |
| | | | 52.80% | 48.80% | 50.70% | |
| | yes | n | 528 | 655 | 1183 | |
| | | | 47.20% | 51.20% | 49.30% | |
| AFT | below 13 | n | 337 | 356 | 693 | 0.209 |
| | | | 30.10% | 27.80% | 28.90% | |
| | above 13 | n | 781 | 924 | 1705 | |
| | | | 69.90% | 72.20% | 71.10% | |
| DSST | below 34 | n | 294 | 317 | 611 | 0.391 |
| | | | 26.30% | 24.80% | 25.50% | |
| | above 34 | n | 824 | 963 | 1787 | |
| | | | 73.70% | 75.20% | 74.50% | |
| IRT | below | n | 374 | 280 | 654 | <0.001 |
| | | | 33.50% | 21.90% | 27.30% | |
| | above | n | 744 | 1000 | 1744 | |
| | | | 66.50% | 78.10% | 72.70% | |
| DRT | below | n | 495 | 449 | 944 | <0.001 |
| | | | 44.30% | 35.10% | 39.40% | |
| | | n | 623 | 831 | 1454 | |
| | | | 55.70% | 64.90% | 60.60% | |

the AFT, there was no significant association between arthritis and cognitive impairment across all three models (Model 1: OR 0.686, 95% CI -0.565 to 0.272; Model 2: OR 0.931, 95% CI -0.614 to 0.219; Model 3: OR 0.446, 95% CI -0.526 to 0.334). In contrast, the DSST showed a significant positive association in Models 1 and 2 (Model 1: OR 1.262, 95% CI -1.954 to 0.424; Model 2: OR 1.538, 95% CI -2.101 to 0.254), but not in Model 3 (OR 0.879, 95% CI -1.745 to 0.665). The CREAD consistently indicated a significant positive association across all three models (Model 1: OR 1.144, 95% CI -0.297 to 0.078; Model 2: OR 0.318, 95% CI -0.573 to 0.413; Model 3: OR 0.214, 95% CI -0.565 to 0.454), suggesting a stronger link between arthritis and cognitive impairment with this assessment tool.

Fig 2 illustrates a scatter matrix plot designed to investigate the potential nonlinear interrelations among the variables once translated into continuous data forms—specifically AFT, DSST, IRT, DRT and age. This graphical representation serves to probe the intricacies beyond linear associations within the dataset.

Fig 3 conveys the outcomes of the binary logistic regression analysis assessing the impact of arthritis on cognitive proficiency. The analysis, which accounted meticulously for an array of sociodemographic attributes, behavioral patterns, and health-related variables, offers a nuanced interpretation. Older adults with arthritis had slightly fewer words in delayed word recall compared to older adults without arthritis, but this association was not significant in the adjusted model (p = 0.134, p = 0.247, p = 0.292). After further adjustment for sociodemographic, behavioral, and health factors, it was found that older adults with arthritis performed significantly worse on the AFT and DSST compared to older adults without arthritis (AFT: p = 0.02, p = 0.018, p = 0.028; DSST: p = 0.027, p = 0.021, p = 0.025).

**Table 2. Characteristics of participants with/without the cognitive impairment.**

| Variables | Classification | N | AFT | | p | DSST | | p | IRT | | p | DRT | | p |
|---|---|---|---|---|---|---|---|---|---|---|---|---|---|---|
| | | | no | yes | | no | yes | | no | yes | | no | yes | |
| Age | 60–69 | n | 326 | 1010 | <0.001 | 272 | 1064 | <0.001 | 267 | 1069 | <0.001 | 416 | 920 | <0.001 |
| | | | 47.00% | 59.20% | | 44.50% | 59.50% | | 40.80% | 61.30% | | 44.10% | 63.30% | |
| | 70–80 | n | 367 | 695 | | 339 | 723 | | 387 | 675 | | 528 | 534 | |
| | | | 53.00% | 40.80% | | 55.50% | 40.50% | | 59.20% | 38.70% | | 55.90% | 36.70% | |
| Gender | male | n | 328 | 832 | 0.514 | 341 | 819 | <0.001 | 382 | 778 | <0.001 | 544 | 616 | <0.001 |
| | | | 47.30% | 48.80% | | 55.80% | 45.80% | | 58.40% | 44.60% | | 57.60% | 42.40% | |
| | female | n | 365 | 873 | | 270 | 968 | | 272 | 966 | | 400 | 838 | |
| | | | 52.70% | 51.20% | | 44.20% | 54.20% | | 41.60% | 55.40% | | 42.40% | 57.60% | |
| Race | Mexican American | n | 54 | 151 | <0.001 | 82 | 123 | <0.001 | 69 | 136 | 0.002 | 89 | 116 | <0.001 |
| | | | 7.80% | 8.90% | | 13.40% | 6.90% | | 10.60% | 7.80% | | 9.40% | 8.00% | |
| | Other Hispanic | n | 85 | 153 | | 113 | 125 | | 85 | 153 | | 111 | 127 | |
| | | | 12.30% | 9.00% | | 18.50% | 7.00% | | 13.00% | 8.80% | | 11.80% | 8.70% | |
| | Non-Hispanic White | n | 239 | 940 | | 182 | 997 | | 291 | 888 | | 448 | 731 | |
| | | | 34.50% | 55.10% | | 29.80% | 55.80% | | 44.50% | 50.90% | | 47.50% | 50.30% | |
| | Non-Hispanic Black | n | 230 | 323 | | 199 | 354 | | 149 | 404 | | 232 | 321 | |
| | | | 33.20% | 18.90% | | 32.60% | 19.80% | | 22.80% | 23.20% | | 24.60% | 22.10% | |
| | Other Race | n | 85 | 138 | | 35 | 188 | | 60 | 163 | | 64 | 159 | |
| | | | 12.30% | 8.10% | | 5.70% | 10.50% | | 9.20% | 9.30% | | 6.80% | 10.90% | |
| Marriage | Married | n | 356 | 964 | <0.001 | 289 | 1031 | <0.001 | 337 | 983 | <0.001 | 499 | 821 | 0.002 |
| | | | 51.40% | 56.50% | | 47.30% | 57.70% | | 51.50% | 56.40% | | 52.90% | 56.50% | |
| | Widowed | n | 171 | 284 | | 162 | 293 | | 162 | 293 | | 214 | 241 | |
| | | | 24.70% | 16.70% | | 26.50% | 16.40% | | 24.80% | 16.80% | | 22.70% | 16.60% | |
| | Divorced | n | 92 | 269 | | 75 | 286 | | 75 | 286 | | 125 | 236 | |
| | | | 13.30% | 15.80% | | 12.30% | 16.00% | | 11.50% | 16.40% | | 13.20% | 16.20% | |
| | Separated | n | 28 | 38 | | 36 | 30 | | 22 | 44 | | 29 | 37 | |
| | | | 4.00% | 2.20% | | 5.90% | 1.70% | | 3.40% | 2.50% | | 3.10% | 2.50% | |
| | Never married | n | 33 | 101 | | 32 | 102 | | 35 | 99 | | 48 | 86 | |
| | | | 4.80% | 5.90% | | 5.20% | 5.70% | | 5.40% | 5.70% | | 5.10% | 5.90% | |
| | Living with partner | n | 13 | 49 | | 17 | 45 | | 23 | 39 | | 29 | 33 | |
| | | | 1.90% | 2.90% | | 2.80% | 2.50% | | 3.50% | 2.20% | | 3.10% | 2.30% | |
| Diabetes | yes | n | 200 | 388 | 0.002 | 202 | 386 | <0.001 | 189 | 399 | 0.002 | 258 | 330 | 0.01 |
| | | | 28.90% | 22.80% | | 33.10% | 21.60% | | 28.90% | 22.90% | | 27.30% | 22.70% | |
| | no | n | 493 | 1317 | | 409 | 1401 | | 465 | 1345 | | 686 | 1124 | |
| | | | 71.10% | 77.20% | | 66.90% | 78.40% | | 71.10% | 77.10% | | 72.70% | 77.30% | |
| Drink | yes | n | 431 | 1214 | <0.001 | 377 | 1268 | <0.001 | 428 | 1217 | 0.041 | 640 | 1005 | 0.495 |
| | | | 62.20% | 71.20% | | 61.70% | 71.00% | | 65.40% | 69.80% | | 67.80% | 69.10% | |
| | no | n | 262 | 491 | | 234 | 519 | | 226 | 527 | | 304 | 449 | |
| | | | 37.80% | 28.80% | | 38.30% | 29.00% | | 34.60% | 30.20% | | 32.20% | 30.90% | |
| Exercise | yes | n | 139 | 530 | <0.001 | 118 | 551 | <0.001 | 147 | 522 | <0.001 | 221 | 448 | <0.001 |
| | | | 20.10% | 31.10% | | 19.30% | 30.80% | | 22.50% | 29.90% | | 23.40% | 30.80% | |
| | no | n | 554 | 1175 | | 493 | 1236 | | 507 | 1222 | | 723 | 1006 | |
| | | | 79.90% | 68.90% | | 80.70% | 69.20% | | 77.50% | 70.10% | | 76.60% | 69.20% | |
| Hypertension | yes | n | 470 | 1014 | <0.001 | 430 | 1054 | <0.001 | 430 | 1054 | 0.017 | 604 | 880 | 0.088 |
| | | | 67.80% | 59.50% | | 70.40% | 59.00% | | 65.70% | 60.40% | | 64.00% | 60.50% | |
| | no | n | 223 | 691 | | 181 | 733 | | 224 | 690 | | 340 | 574 | |
| | | | 32.20% | 40.50% | | 29.60% | 41.00% | | 34.30% | 39.60% | | 36.00% | 39.50% | |

(*Continued*)

**Table 2.** (Continued)

| Variables | Classification | N | AFT | | p | DSST | | p | IRT | | p | DRT | | p |
|---|---|---|---|---|---|---|---|---|---|---|---|---|---|---|
| | | | no | yes | | no | yes | | no | yes | | no | yes | |
| Arthritis | no | n | 327 | 888 | 0.03 | 292 | 923 | 0.099 | 329 | 886 | 0.828 | 493 | 722 | 0.219 |
| | | | 47.20% | 52.10% | | 47.80% | 51.70% | | 50.30% | 50.80% | | 52.20% | 49.70% | |
| | yes | n | 366 | 817 | | 319 | 864 | | 325 | 858 | | 451 | 732 | |
| | | | 52.80% | 47.90% | | 52.20% | 48.30% | | 49.70% | 49.20% | | 47.80% | 50.30% | |
| Smoke | yes | n | 351 | 872 | 0.826 | 320 | 903 | 0.432 | 320 | 903 | 0.214 | 499 | 724 | 0.142 |
| | | | 50.60% | 51.10% | | 52.40% | 50.50% | | 48.90% | 51.80% | | 52.90% | 49.80% | |
| | no | n | 342 | 833 | | 291 | 884 | | 334 | 841 | | 445 | 730 | |
| | | | 49.40% | 48.90% | | 47.60% | 49.50% | | 51.10% | 48.20% | | 47.10% | 50.20% | |
| Sleep | yes | n | 211 | 508 | 0.752 | 176 | 543 | 0.462 | 182 | 537 | 0.158 | 270 | 449 | 0.234 |
| | | | 30.40% | 29.80% | | 28.80% | 30.40% | | 27.80% | 30.80% | | 28.60% | 30.90% | |
| | no | n | 482 | 1197 | | 435 | 1244 | | 472 | 1207 | | 674 | 1005 | |
| | | | 69.60% | 70.20% | | 71.20% | 69.60% | | 72.20% | 69.20% | | 71.40% | 69.10% | |
| BMI | no | n | 147 | 307 | 0.069 | 128 | 326 | 0.14 | 127 | 327 | 0.71 | 185 | 269 | 0.503 |
| | | | 21.20% | 18.00% | | 20.90% | 18.20% | | 19.40% | 18.80% | | 19.60% | 18.50% | |
| | yes | n | 546 | 1398 | | 483 | 1461 | | 527 | 1417 | | 759 | 1185 | |
| | | | 78.80% | 82.00% | | 79.10% | 81.80% | | 80.60% | 81.30% | | 80.40% | 81.50% | |
| Income | no | n | 616 | 1324 | <0.001 | 577 | 1363 | <0.001 | 573 | 1367 | <0.001 | 812 | 1128 | <0.001 |
| | | | 88.90% | 77.70% | | 94.40% | 76.30% | | 87.60% | 78.40% | | 86.00% | 77.60% | |
| | yes | n | 77 | 381 | | 34 | 424 | | 81 | 377 | | 132 | 326 | |
| | | | 11.10% | 22.30% | | 5.60% | 23.70% | | 12.40% | 21.60% | | 14.00% | 22.40% | |
| Educ | no | n | 455 | 677 | <0.001 | 471 | 661 | <0.001 | 410 | 722 | <0.001 | 542 | 590 | <0.001 |
| | | | 65.70% | 39.70% | | 77.10% | 37.00% | | 62.70% | 41.40% | | 57.40% | 40.60% | |
| | yes | n | 238 | 1028 | | 140 | 1126 | | 244 | 1022 | | 402 | 864 | |
| | | | 34.30% | 60.30% | | 22.90% | 63.00% | | 37.30% | 58.60% | | 42.60% | 59.40% | |

## 4 Discussion

In the current investigation, drawing upon data from NHANES, we probed the association between arthritis and cognitive function in the elder demographic. It was observed that older individuals with arthritis had an increased propensity for cognitive deficits. The outcomes derived from the multiple linear regression analysis, alongside the scatter matrix plots, indicated an absence of a linear correlation between the presence of arthritis and the augmented

**Table 3. Multiple linear regression associations of arthritis with cognitive impairment in American adults.**

| var | 1 | classification | model1 | | | model2 | | | model3 | | |
|---|---|---|---|---|---|---|---|---|---|---|---|
| | | | t | 95%CI | | t | 95%CI | | t | 95%CI | |
| | | | | LL | UL | | LL | UL | | LL | UL |
| AFT | arthritis | no | ref | ref | ref | ref | ref | ref | ref | ref | ref |
| | | yes | -0.686 | -0.565 | 0.272 | -0.931 | -0.614 | 0.219 | -0.44 | -0.526 | 0.334 |
| DSST | arthritis | no | ref | ref | ref | ref | ref | ref | ref | ref | ref |
| | | yes | -1.262 | -1.954 | 0.424 | -1.538 | -2.101 | 0.254 | -0.879 | -1.745 | 0.665 |
| IRT | arthritis | no | ref | ref | ref | ref | ref | ref | ref | ref | ref |
| | | yes | -0.913 | -0.512 | 0.218 | -0.318 | -0.516 | 0.191 | -0.656 | -0.47 | 0.235 |
| DRT | arthritis | no | ref | ref | ref | ref | ref | ref | ref | ref | ref |
| | | yes | 0.983 | -0.091 | 0.274 | 0.914 | -0.095 | 0.26 | 1.092 | -0.078 | 0.275 |

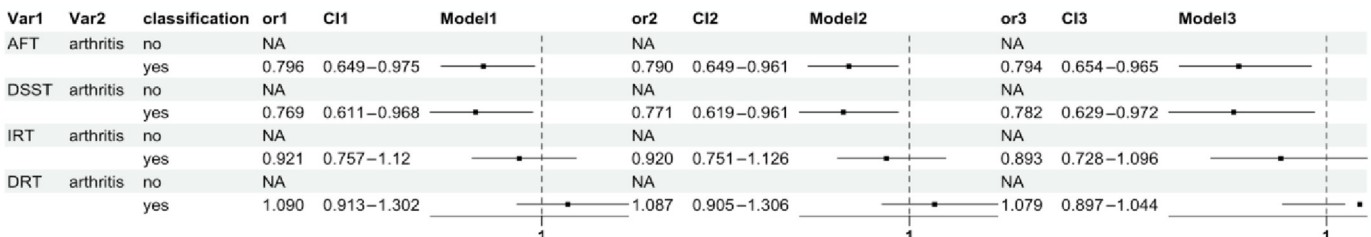

**Fig 2. Scatter matrix plot (converting AFT, DSST, CREAD and age to continuous variables).**

risk of cognitive decline. Logistic regression analyses elucidated that arthritis bore a substantial correlation with cognitive performance as evidenced by AFT and DSST scores, although this association was not mirrored in CERAD scores. Remarkably, even upon rigorous adjustment for potential confounders, arthritis maintained a significant and negative correlation with

| Var1 | Var2 | classification | or1 | CI1 | Model1 | or2 | CI2 | Model2 | or3 | CI3 | Model3 |
|------|------|----------------|-----|-----|--------|-----|-----|--------|-----|-----|--------|
| AFT | arthritis | no | NA | | | NA | | | NA | | |
| | | yes | 0.796 | 0.649–0.975 | | 0.790 | 0.649–0.961 | | 0.794 | 0.654–0.965 | |
| DSST | arthritis | no | NA | | | NA | | | NA | | |
| | | yes | 0.769 | 0.611–0.968 | | 0.771 | 0.619–0.961 | | 0.782 | 0.629–0.972 | |
| IRT | arthritis | no | NA | | | NA | | | NA | | |
| | | yes | 0.921 | 0.757–1.12 | | 0.920 | 0.751–1.126 | | 0.893 | 0.728–1.096 | |
| DRT | arthritis | no | NA | | | NA | | | NA | | |
| | | yes | 1.090 | 0.913–1.302 | | 1.087 | 0.905–1.306 | | 1.079 | 0.897–1.044 | |

**Fig 3. Forest plots of multiple logistic regression.**

enhanced cognitive scores in Model 3 (OR: 0.796, 95% CI: 0.649–0.975; OR: 0.769, 95% CI: 0.611–0.968). Hence, the study posits that arthritis is a potential risk factor for the advent of cognitive impairment within the elder in the United States.

Arthritis is a chronic inflammatory disease. Research has shown that chronic inflammation may affect cognitive function through multiple pathways [28–30]. For example, inflammatory factors such as tumor necrosis factor (TNF) and interleukin-6 (IL-6) are able to cross the blood-brain barrier and directly affect brain cell function [30]. In addition, prolonged inflammatory responses may lead to neuroinflammation and neurodegenerative pathologies, which can cause cognitive decline [28,29]. Within a 20-year longitudinal cohort study employing the OLINK Proteomics Inflammation Panel, researchers have identified that a suite of inflammatory protein biomarkers—including Interleukin 10 (IL10), Leukemia Inhibitory Factor Receptor (LIF-R), Chemokine (C-C motif) Ligand 19 (CCL19), Interleukin 17C (IL-17C), Monocyte Chemoattractant Protein 4 (MCP-4), and Transforming Growth Factor Alpha (TGF-α)—exhibits an association with the trajectory of cognitive deterioration and the subsequent risk of onset of Alzheimer's disease (AD) [31]. Many current studies have reported a complex role for inflammation in dementia-related phenotypes [32–35]. Luo J and colleagues report a possible causal relationship between high IL-8 levels and better cognitive performance but smaller hippocampal volumes in the general healthy population [35]. Chen BA et al. found that higher serum fibrinogen levels were negatively associated with cognitive functioning [32].

Prior research has yielded varied yet statistically significant correlations between demographic factors or characteristics of arthritis and cognitive assessments [36]. Concordant outcomes were echoed in subsequent studies [37–40], which align with the observations presented in our findings. However, results from a study [41] of 4,462 participants interviewed in the nationally representative U.S. Health and Retirement Study with linked Medicare claims in 2016 showed no association between RA and CI in this national sample of older Americans. But a cross-sectional study [42] from China reported that participants with arthritis had a higher risk of cognitive impairment compared with those without arthritis after adjusting for sociodemographic, lifestyle behavior, and mental health conditions (OR = 1.322, 95% CI: 1.022–1.709), which is also consistent with our results. And a recent Mendelian randomization study showed genetic support for a causal relationship between RA and increased risk of cognitive impairment [43]. At present, the two different conclusions are still not comparable, and we consider that this is due to the limitations of the cross-sectional study, the under-representation of the sample data from different countries, and that an increase in the sample size and the use of sample weights may lead to more accurate results.

An investigation involving 1,449 participants revealed a robust association between the emergence of joint diseases in midlife and a subsequent decline in cognitive function over the span of 21 years [44]. Complementary findings from another longitudinal study indicate that individuals with arthritis exhibit diminished cognitive performance compared to their non-arthritic counterparts, as evidenced by lower performance on tests of situational memory, mental status, and overall cognitive ability [45]. Moreover, recent Mendelian randomization studies present divergent conclusions regarding the putative causal link between RA and AD [46,47]. In an animal model representing RA, an upsurge in aberrant astrocytic production of gamma-Aminobutyric acid (GABA), which is mediated by Monoamine Oxidase B (MAO-B), is potentiated by leukocyte interleukin-1β. This biochemical activity leads to the inhibition of CA1 hippocampal pyramidal neurons that play a pivotal role in the consolidation and storage of memory [40]. Skeletal disorders exert a significant influence on both motor skills and cognitive capabilities in the older populations [48]. OA is implicated in an elevated risk for developing dementia and cognitive deficits, suggesting that efficacious interventions targeting OA patients may mitigate the onset rate of both dementia and cognitive impairments [17].

Induction of osteoarthritis led to a decline in cognitive capabilities as assessed by the Novel Object Recognition Test (NORT) in rats [49]. Variations in mood and cognitive functions were distinctly observed post-induction of joint pain in kappa-opioid receptor knockout (KOR-KO) and prodynorphin knockout (PDYN-KO) mice [50].

In addition to the impairment of cognitive function that occurs as a result of the chronic inflammatory response that accompanies arthritis, several related mechanisms are involved. Patients with arthritis are often at increased risk for cardiovascular diseases, such as atherosclerosis and hypertension, and these cardiovascular problems are strongly associated with cognitive decline [43,51]. Cardiovascular disease can lead to reduced cerebral blood flow and impair brain function, which can affect cognitive performance [52]. In addition, treatment of arthritis often requires long-term use of immunosuppressive and anti-inflammatory medications, which may negatively affect cognitive function [53–55]. For example, long-term use of glucocorticoids has been shown to be associated with cognitive dysfunction [55]. Furthermore, arthritis is often associated with chronic pain and dysfunction, and these factors can lead to depression, anxiety, and decreased quality of life, which can indirectly affect cognitive performance [56–58]. Depression and anxiety themselves have been shown to be associated with cognitive decline [42,59]. Although the mechanisms that lead to the association between arthritis and cognitive impairment remain uncertain, the implementation of preventive strategies, such as regular assessment of cognitive function in individuals diagnosed with arthritis, is recommended in professional practice. This is an important study not only for the older adults of life of elderly people with arthritis, but also for the harmonious living of families and communities.

Nonetheless, certain constraints within our approach warrant acknowledgment. Based on multiple linear regression analyses, the association between arthritis and cognitive impairment in U.S. adults varied depending on the cognitive assessment tool used. When assessed using the CERAD scale, arthritis was significantly associated with higher cognitive dysfunction in all three models, as evidenced by odds ratios and confidence intervals. However, when using the AFT and DSST scales, the relationship between arthritis and cognitive impairment was inconsistent, significant in some models and not in others. These findings underscore the importance of assessment tools in measuring cognitive impairment and its relationship with arthritis, suggesting that larger sample sizes and multicenter studies are needed for further exploration. The cross-sectional essence of our study intrinsically includes confounding factors and hinders the definitive ascertainment of causative ties between arthritis and cognitive function impairment. Moreover, the scope of analysis was delineated to NHANES questionnaire data collected between 2011 and 2014, thus not facilitating an investigation into the nexus between arthritis and particularized clinical expressions of cognitive function impairment, such as visuospatial impairment and implementation difficulties. Furthermore, our study was based exclusively on European and American patients, and due to the expected heterogeneity and propensity for multiple effects, our conclusions need to be validated with a wider range of data.

## 5 Conclusion

Scores from AFT and DSST were significantly elevated in individuals diagnosed with arthritis when compared to non-arthritic counterparts; however, no such increase was observed in IRT and DRT levels. The linkage of arthritis with defects in specific cognitive functions was notably patent. These findings align with extant literature positing an association between arthritis and cognitive decline. Prospective investigations, encompassing both mechanistic and longitudinal studies, are crucial to delve into the etiology of this distinct association. Moreover, such

research is imperative to delineate the direct impact of arthritis on cognitive functioning and devise strategies to circumscribe the limitations presented by this study.

## Acknowledgments

Our profound gratitude is extended to the committed team at the Centers for Disease Control and Prevention (CDC), as well as the personnel at the National Center for Health Statistics (NCHS), for their indispensable support. We further wish to acknowledge the contributions of the National Health and Nutrition Examination Survey participants, whose participation has been pivotal to the success of this study.

## Author Contributions

**Conceptualization:** Taihong Lv.

**Data curation:** Taihong Lv, Hanming Yu.

**Formal analysis:** Taihong Lv.

**Investigation:** Hanming Yu.

**Methodology:** Hanming Yu.

**Project administration:** Li Ma.

**Resources:** Zishuo Ji.

**Software:** Zishuo Ji.

**Writing – original draft:** Taihong Lv.

**Writing – review & editing:** Li Ma.

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
