## [Decision Letter · Decision Letter 0]

19 Jul 2024

PONE-D-24-26744The Association Among Arthritis and Cognitive Function Impairment in the Elderly: Based on the NHANES 2011-2014PLOS ONE

Dear Dr. Ma,

Thank you for submitting your manuscript to PLOS ONE. After careful consideration, we feel that it has merit but does not fully meet PLOS ONE’s publication criteria as it currently stands. Therefore, we invite you to submit a revised version of the manuscript that addresses the points raised during the review process.

We look forward to receiving your revised manuscript.

Kind regards,

Jaspinder Kaur, MD

Academic Editor

PLOS ONE

Journal Requirements:

Reviewers' comments:

Reviewer's Responses to Questions

**Comments to the Author**

1. Is the manuscript technically sound, and do the data support the conclusions?

Reviewer #1: Yes

Reviewer #2: No

2. Has the statistical analysis been performed appropriately and rigorously? 

Reviewer #1: I Don't Know

Reviewer #2: No

3. Have the authors made all data underlying the findings in their manuscript fully available?

Reviewer #1: Yes

Reviewer #2: Yes

4. Is the manuscript presented in an intelligible fashion and written in standard English?

Reviewer #1: Yes

Reviewer #2: No

5. Review Comments to the Author

Reviewer #1: Ideally, you have analysed an extensive database for correlations between arthritis and dementia.

I suggest minor revisions to this paper.

The author should explain the Abbreviations used in the abstract.

The author should explain the Abbreviations used in the table below for better and faster interpretation of the data.

Reviewer #2: Taihong Lv and co-authors utilized Health and Nutrition Examination Survey (NHANES) from 2011–2012 and 2013–2014 to mine the data regarding Arthritis and its correlation with cognitive decline.

This reviewer feels the manuscript had interesting question to explore about how arthritis could affect cognitive abilities. However, the overall manuscript suffered from multiple competing but incomplete stories. This manuscript has many flaws in English and grammar. Many major mistakes in writing, for instance: Consortium to Establish a Registry for Alzheimer’s disease (CERAD) is abbreviated as CREAD in tables. Authors are confused about the subject matter; they mentioned CERAD W-L is a 3-diethyl-carbamoyl benzoic acid.

Major Concerns:

1) The primary concern is about the methodology utilized in the current study to conclude the correlation. To the best of this reviewer’s understanding, the Consortium to Establish a Registry for Alzheimer’s disease (CERAD) should be including tests that measure word-related learning (CERAD-IR) and memory components (CERAD-DR). Reviewer would like to see more specific details for other parameters like Word List Learning Test (WLLT), the Word List Recall Test (WLRT), and Intrusion Word Count Test (WLLT-IC and WLRT-IC).

2) Reviewer have concerns about the analysis and presentation of results in particular. Overall, the way that the many statistical analyses were presented are somewhat confusing, and I believe this data need to be present in a clear format for a general reader. There need to have a graphical presentation of the major findings about Logistic relationship between cognitive function and arthritis after the adjustment for sociodemographic factors.

3) How about depression and including Patient Health Questionnaire-9 (PHQ-9) in your data mining and correlates that with Arthritis?

4) Results section need to rewrite, with section headings with clear interpretation of particular findings.

Minor Concerns:

1) In Methods: Please expand your Statistical analysis approach in detail.

2) Don’t use the words that may have negative connotations, such as “the aged,” “elderly,” “senior,” “senior citizen,” rather use more polite words “older adults,” “older populations,” Please change it throughout the manuscript.

3) Discussion need to be expand and please propose the mechanism in depth about the correlation of arthritis and cognitive decline by citing recent literature.

4) Study published in 2021 on the same subject that describe that No association between rheumatoid arthritis and cognitive impairment in a cross-sectional national sample of older U.S. adults. PMID: 34404491. Please discuss this in your discussion.

5) Explain how this data and conclusions from your study could be useful in clinic.

6) Describe the limitations of your study and data mining in the discussion section.

6. PLOS authors have the option to publish the peer review history of their article (what does this mean?). If published, this will include your full peer review and any attached files.

Reviewer #1: **Yes: **Assoc. Prof. Robert Olszewski, MD, PhD, FESC

Reviewer #2: No

---

## [Author Response · Author response to Decision Letter 0]

30 Jul 2024

We would like to thank the reviewers for reviewing our manuscript, and we have revised it according to their suggestions, as detailed in the ‘Response to Reviewers’ document.

---

## [Decision Letter · Decision Letter 1]

15 Aug 2024

PONE-D-24-26744R1The Association Among Arthritis and Cognitive Function Impairment in the Elderly: Based on the NHANES 2011-2014PLOS ONE

Dear Dr. Ma,

Thank you for submitting your manuscript to PLOS ONE.  Well done on your article. There are minor word changes that needs to be addressed as suggested by reviewers. Please do the needful and submit the final copy.

We look forward to receiving your revised manuscript.

Kind regards,

Jaspinder Kaur, MD

Academic Editor

PLOS ONE

Journal Requirements:

Reviewers' comments:

Reviewer's Responses to Questions

**Comments to the Author**

1. If the authors have adequately addressed your comments raised in a previous round of review and you feel that this manuscript is now acceptable for publication, you may indicate that here to bypass the “Comments to the Author” section, enter your conflict of interest statement in the “Confidential to Editor” section, and submit your "Accept" recommendation.

Reviewer #2: All comments have been addressed

Reviewer #3: All comments have been addressed

2. Is the manuscript technically sound, and do the data support the conclusions?

Reviewer #2: Yes

Reviewer #3: Yes

3. Has the statistical analysis been performed appropriately and rigorously? 

Reviewer #2: Yes

Reviewer #3: Yes

4. Have the authors made all data underlying the findings in their manuscript fully available?

Reviewer #2: Yes

Reviewer #3: Yes

5. Is the manuscript presented in an intelligible fashion and written in standard English?

Reviewer #2: Yes

Reviewer #3: Yes

6. Review Comments to the Author

Reviewer #2: Congratulations for incorporating all the suggestions.

Please make these minor changes:

Change elderly to "older adults" in these sentences:

1) In your title, "Line 2 Impairment in the Elderly: Based on the NHANES 2011-2014"

2) Abstract, line 37, elders.

2) Line 86 impairment incidence among the elderly in the United States.

3) Line 300 study not only for the personal quality of life of elderly people with arthritis

Reviewer #3: the authors have corrected all suggestion from the previous round of review

the manuscript is sound and the data supports the conclusion

i think the statistical analysis is done well

yes the authors have made all the data/findings available

the language of the manuscript is of good scientific level

minor changes/suggestions

line 1 title between instead of among

line 28 were instead of was

line 29 patients instead of subjects

line 44 were instead of are

line 269 Alzheimer disease abbreviation repeated (AD)

line 319 patients instead of subjects

7. PLOS authors have the option to publish the peer review history of their article (what does this mean?). If published, this will include your full peer review and any attached files.

Reviewer #2: No

Reviewer #3: **Yes: **mohammad al-magableh

---

## [Author Response · Author response to Decision Letter 1]

22 Aug 2024

We are grateful to the editors and reviewers for their review of our manuscript, and we have responded in detail to the revisions in attachment ‘Response to Reviewers’.

---

## [Decision Letter · Decision Letter 2]

3 Sep 2024

The Association Among Arthritis and Cognitive Function Impairment in the older adults: Based on the NHANES 2011-2014

PONE-D-24-26744R2

Dear Dr. Ma,

We’re pleased to inform you that your manuscript has been judged scientifically suitable for publication and will be formally accepted for publication once it meets all outstanding technical requirements.

Kind regards,

Zhe He, PhD

Academic Editor

PLOS ONE

Additional Editor Comments (optional):

The authors have addressed comments of both reviewers.

Reviewers' comments:

Reviewer's Responses to Questions

**Comments to the Author**

1. If the authors have adequately addressed your comments raised in a previous round of review and you feel that this manuscript is now acceptable for publication, you may indicate that here to bypass the “Comments to the Author” section, enter your conflict of interest statement in the “Confidential to Editor” section, and submit your "Accept" recommendation.

Reviewer #2: All comments have been addressed

Reviewer #3: All comments have been addressed

2. Is the manuscript technically sound, and do the data support the conclusions?

Reviewer #2: Yes

Reviewer #3: Yes

3. Has the statistical analysis been performed appropriately and rigorously? 

Reviewer #2: Yes

Reviewer #3: Yes

4. Have the authors made all data underlying the findings in their manuscript fully available?

Reviewer #2: Yes

Reviewer #3: Yes

5. Is the manuscript presented in an intelligible fashion and written in standard English?

Reviewer #2: Yes

Reviewer #3: Yes

6. Review Comments to the Author

Reviewer #2: (No Response)

Reviewer #3: all the comments have been done and corrected in a good scientific manner, the authors have made a good change to the article

7. PLOS authors have the option to publish the peer review history of their article (what does this mean?). If published, this will include your full peer review and any attached files.

Reviewer #2: **Yes: **Dr. Pankaj Patyal, PhD

Reviewer #3: **Yes: **mohammad al-magableh

---

## [Editor Report · Acceptance letter]

19 Sep 2024

PONE-D-24-26744R2 

PLOS ONE

Dear Dr. Ma, 

I'm pleased to inform you that your manuscript has been deemed suitable for publication in PLOS ONE. Congratulations! Your manuscript is now being handed over to our production team.

Kind regards, 

on behalf of

Dr. Zhe He 

Academic Editor

PLOS ONE